# Triarylpyridine Compounds and Chloroquine Act in Concert to Trigger Lysosomal Membrane Permeabilization and Cell Death in Cancer Cells

**DOI:** 10.3390/cancers12061621

**Published:** 2020-06-18

**Authors:** Jennifer Beauvarlet, Rabindra Nath Das, Karla Alvarez-Valadez, Isabelle Martins, Alexandra Muller, Elodie Darbo, Elodie Richard, Pierre Soubeyran, Guido Kroemer, Jean Guillon, Jean-Louis Mergny, Mojgan Djavaheri-Mergny

**Affiliations:** 1Institut Bergonié, INSERM U1218, Université de Bordeaux, 33000 Bordeaux, France; beauvarletj@gmail.com (J.B.); elodie.darbo@gmail.com (E.D.); robertelo@yahoo.fr (E.R.); P.Soubeyran@bordeaux.unicancer.fr (P.S.); 2ARNA Laboratory, Université de Bordeaux, INSERM U1212, CNRS UMR 5320, 33000 Bordeaux, France; rabindrachem@gmail.com (R.N.D.); jean.guillon@u-bordeaux.fr (J.G.); jean-louis.mergny@inserm.fr (J.-L.M.); 3Metabolomics and Cell Biology Platforms, Institut Gustave Roussy, 94805 Villejuif, France; alvarezvaladez4@gmail.com (K.A.-V.); alexandra.muller12@gmail.com (A.M.); kroemer@orange.fr (G.K.); 4Centre de Recherche des Cordeliers, INSERM UMRS 1138, Sorbonne Université, Université de Paris, Equipe 11 labellisée par la Ligue contre le Cancer, 75006 Paris, France; 5Gustave Roussy Comprehensive Cancer Institute, 94805 Villejuif, France; isabellemart@gmail.com; 6Pôle de Biologie, Hôpital Européen Georges Pompidou, AP-HP, 75015 Paris, France; 7Suzhou Institute for Systems Medicine, Chinese Academy of Sciences, Suzhou 215123, China; 8Department of Women’s and Children’s Health, Karolinska University Hospital, 17176 Stockholm, Sweden

**Keywords:** Lysosome, lysosomal membrane permeabilization, cell death, cancer, G-quadruplex ligand, triarylpyridine compounds, resistance to therapy

## Abstract

Lysosomes play a key role in regulating cell death in response to cancer therapies, yet little is known on the possible role of lysosomes in the therapeutic efficacy of G-quadruplex DNA ligands (G4L) in cancer cells. Here, we investigate the relationship between the modulation of lysosomal membrane damage and the degree to which cancer cells respond to the cytotoxic effects of G-quadruplex ligands belonging to the triarylpyridine family. Our results reveal that the lead compound of this family, **20A** promotes the enlargement of the lysosome compartment as well as the induction of lysosome-relevant mRNAs. Interestingly, the combination of **20A** and chloroquine (an inhibitor of lysosomal functions) led to a significant induction of lysosomal membrane permeabilization coupled to massive cell death. Similar effects were observed when chloroquine was added to three new triarylpyridine derivatives. Our findings thus uncover the lysosomal effects of triarylpyridines compounds and delineate a rationale for combining these compounds with chloroquine to increase their anticancer effects.

## 1. Introduction

Lysosomes are the end points of several vesicular pathways that mediate degradation of intra- and extra-cellular material in cells [1]. As a result, lysosomes ensure the regulation of numerous biological processes including nutrient sensing, signaling to the nucleus, and the ignition of cell death pathways [2]. In particular, lysosomes are essential for the proper function of autophagy, a cytoprotective process that provides necessary nutrients and energy through promoting the degradation of intracellular material [3,4]. Hallmarks of lysosomal dysfunction include changes in lysosomal size, morphology and localization, changes in the expression of lysosomal enzymes and alterations of lysosomal membrane permeability [5].

Lysosomes play major roles in cancer, as they undergo multiple alterations during the transformation process that contribute to tumor progression and metastasis [6]. Mounting evidence indicates that lysosomes are involved in the resistance to anticancer compounds with weak base properties. These drugs may be sequestered within the lysosome, resulting in a loss of accessibility of the drug to its cellular targets and thereby a reduction of its effectiveness [7,8,9,10]. The mechanism of lysosomal sequestration can be passive, through diffusion into the acidic lumen of the lysosome where the drug is protonated and retained [10,11]. However, an active mechanism has also been described that relies on multidrug efflux transporters [12]. Accordingly, inhibition of lysosomal sequestration (by increasing lysosomal pH, or inhibiting active transport) may increase the sensitivity of cancer cells to several antineoplastic therapies [13]. Conversely, the modifications of the lysosomal compartment that occur during cellular transformation can paradoxically render cancer cells more susceptible to lysosomal membrane permeabilization (LMP) [14,15], and ultimately lead to lysosomal-dependent cell death (LDCD) induced by anticancer agents [13,16].

LDCD is characterized as a process that involves LMP and the subsequent release of lysosomal content such as cathepsins into the cytosol [17]. Extensive lysosomal leakage can cause necrotic cell death while partial lysosomal membrane destabilization can result in lysosomal cell death through apoptotic and non-apoptotic dependent mechanisms. LMP is triggered by stimuli that affect lysosomal membrane stability, including lysosomotropic agents, ROS-generating molecules, or agents that affect the lipid composition of the lysosomal membrane [17]. Of note, the induction of lysosomal-dependent non-apoptotic cell death is of tremendous interest for cancer therapy because cancer cells often display a defective apoptotic machinery rendering them resistant to conventional chemotherapies [18].

Hence, the identification of molecules (or combination of molecules) that promote lysosomal cell death is an attractive strategy for the treatment of apoptosis- or therapy-resistant cancers. Quadruplexes (G4) are unusual nucleic acid structures formed by G-rich DNA and RNA [19] that can be selectively recognized by ligands. A number of G4 ligands have been described and over a thousand different compounds are now listed in the G4 ligand database (http://www.g4ldb.org). These compounds can be neutral or bear one or more positive or negative charges thanks to differences in side chains [20,21,22]; several of these molecules exhibit exquisite selectivity towards G-quadruplexes in vitro. The effect of G4 ligands on genomic stability, replication, transcription and splicing and their roles as anticancer agents is starting to emerge [23], but a detailed analysis of their effects on lysosome is still lacking. We have previously shown that **20A**, a G-quadruplex ligand belonging to the triarylpyridine family, exerts potent cytotoxic and cytostatic effects on several cancer cells. We found that one of the gene classes most significantly upregulated by **20A** corresponds to the autophagy lysosomal pathway [24].

In this work, we speculated that **20A** may interfere with the lysosomal pathway as autophagy and lysosomal function are intertwined processes. A more detailed analysis of the lysosomal KEGG database indeed revealed that **20A** led to an increased expression of numerous genes involved in the lysosomal pathway (i.e., genes coding for lysosomal hydrolases and membrane proteins, as well as for proteins involved in the synthesis and traffic of lysosomal enzymes) (Appendix A). Based on these observations, we hypothesized that the enrichment of the lysosomal pathway by **20A** would render cells less vulnerable to the cytotoxic effect of this compound. We thus investigated the effect of chloroquine, an inhibitor of lysosomal functions, on the cytotoxic effect of G4 ligands from the triarylpyridine family. The chemical structures of triarylpyridine compounds used in this study are shown in Figure 1. In addition to the parent compound **20A**, three new **20A** derivatives were synthesized (Appendix A) and then tested for their ability to stabilize G4 structures. These three new **20A** derivatives stabilize the human telomeric quadruplex in a concentration-dependent manner (Appendix A), as previously reported for **20A** [25].

## 2. Results

### 2.1. ***20A*** Accumulates Within the Lysosomal Compartment and Causes Enlargement of the Lysosomes

To investigate the effect of **20A** on lysosomal homeostasis, we first stained cells with the lysosomal marker, LysoTracker Red. As shown in Figure 2a, the size of the lysosomes is significantly increased following exposure of HeLa cells to **20A**. Incidentally, we observed the presence of faint blue fluorescence puncta in the cytosol of cells subjected to **20A** treatment (Figure 2b). In vitro fluorescence experiments confirmed that **20A** emits a weak blue fluorescence following excitation in the UV region (Appendix A). Co-localization experiments with LysoTracker and MitoTracker (a specific marker of mitochondria) revealed that the blue fluorescence puncta mostly co-localize with the lysosomes, not with mitochondria (Figure 2b,c). Altogether, these results indicate that **20A** accumulates within the lysosome as early as 6 h after treatment and elicits an enlargement of this compartment.

### 2.2. ***20A*** and Chloroquine Act in Concert to Trigger Cell Death

The presence of **20A** within lysosomes is reminiscent of a mechanism referred to as lysosomal drug sequestration, which is observed in response to some anticancer compounds [18]. Growing evidence reveals that lysosomotropic agents such as chloroquine can prevent lysosomal drug sequestration, thus sensitizing cells to cancer therapies [10,11,26]. To test this hypothesis, we investigated the effect of a combination of **20A** and chloroquine on cell death in both HeLa cervical carcinoma and A549 non-small cell lung cancer cell lines. In order to properly evaluate the effect of the combination of the two drugs (**20A** and chloroquine), we decided to use **20A** doses that trigger a relatively low level of cell death (<10%) when used alone. Similarly, a sublethal dose (<10% cell death) of chloroquine (25 µM) was used.

As shown in Figure 3a, the combination of chloroquine (25 µM) and **20A** (5 or 6 µM) promoted a robust activation of cell death (50% or more) in HeLa cells as compared to **20A** or chloroquine alone. A similar effect was observed when A549 cells were treated with the combination of **20A** (3.5 or 4 µM) and chloroquine (25 µM), indicating that chloroquine greatly potentiated **20A**-induced cell death. Similar results were found with another lysosomal inhibitor, Lys05, which is an analogue of chloroquine (data not shown). As found for HeLa and A549 cells, the combination of **20A** and chloroquine promoted a significant cell death induction in U20S osteosarcoma cells (Figure 3b, left panel). We also evaluated loss of mitochondrial transmembrane potential (a feature of early apoptosis) and increased cell membrane permeability (a feature of late apoptosis) by co-staining U2OS cells with TMRM and DAPI dyes. As shown in Figure 3b (right panel) and Appendix A, the combination of **20A** and chloroquine led to early (TMRM^low^/DAPI^Low^) and late cell death (TMRM^Low^/DAPI ^High^) as earlier as 8 h of treatment. A significant increase of both features of apoptosis were observed after 16 h of treatment (Figure 3b, right panel).

As chloroquine is a well-known inhibitor of autophagy and **20A** has been shown to promote autophagy [24], we next explored whether the inhibition of autophagy would be responsible for the robust induction of cell death by the combination of **20A** and chloroquine. To this purpose, we compared cell death induced by **20A** plus chloroquine in autophagy-proficient cells and autophagy-deficient cells (the latter were generated by disruption of two key autophagy genes, ATG5 and ATG7); (Figure 3c, left panel and Appendix A for the whole blots). As shown in Figure 3c (right panel), autophagy-deficient cells exhibited a similar rate of cell death in response to **20A** plus chloroquine as did autophagy-proficient cells, suggesting that chloroquine (at least at 25 µM) sensitizes cells to **20A** treatment through an autophagy-independent mechanism.

The induction of lysosomal membrane permeabilization (LMP) represents one mechanism through which chloroquine can affect lysosomes [27]. Recently, the detection of Galectin 3 puncta at damaged endo-lysosome membranes was shown to be a highly sensitive method to evaluate LMP [28]. To explore this possibility, we used the U20S cell line expressing mCherry-tagged Galectin 3 and counted cells displaying one or more Galectin 3 puncta. As shown in Figure 4 (a1 and a2), the percentage of cells with Galectin 3 puncta was nearly equal in cells subjected to **20A** and in untreated cells (3% and 2%, respectively) but increased to 8 % when cells were exposed to chloroquine. However, the percentage of cells with Galectin 3 puncta massively increased (to 39%) when chloroquine was associated with **20A** treatment. It is worth noting that, under the latter condition, 26% of cells with Gal3 puncta harbored one or two Gal3 puncta and the rest (13%) displayed either 3 or more Gal3 puncta (Figure 4a3), suggesting several instances of endo-lysosomal membrane damage per cell. We next analyzed whether the Gal3 puncta were located at the lysosomal membrane by co-staining cells with LAMP1, a specific marker of the lysosomal membrane. 

As shown in Figure 4b, a significant colocalization was observed when cells were exposed to **20A** plus chloroquine, confirming that Gal3 puncta match to the membrane damage sites at lysosomes. We also examined the size of the lysosomes in cells treated with **20A** plus chloroquine by looking at the pattern of LAMP1 staining. As shown in Figure 4b (zoom LAMP1 staining), **20A** and chloroquine both caused a dramatic lysosomal enlargement as evidenced by LAMP1 staining. More importantly, this phenotype was markedly increased when the two compounds were combined.

This observation, combined with the increased size of the lysosomes in cells treated with **20A** plus chloroquine, suggests that the lysosomal swelling process could be a cause of LMP. Altogether, these results demonstrate that the combination of **20A** and chloroquine causes significant lysosomal membrane permeabilization along cell death induction.

### 2.3. Apoptosis Is Involved in Cell Death Induced by ***20A*** Plus Chloroquine

We next investigated whether apoptosis is implicated in the massive cell death induced by **20A** plus chloroquine. To this purpose, we measured apoptosis-associated parameters such as the appearance of the cleaved form of caspase-3 and its substrate PARP1. As presented in Figure 5a (whole blots are shown in Appendix A), the combination of chloroquine and **20A** treatment promoted a robust cleavage of PARP-1 and Caspase 3 in both HeLa and A549 cells, while no significant cleavage was observed when cells were treated solely with one compound.

To further evaluate the role of caspases in the occurrence of cell death, we used a pan-inhibitor of caspase activity (Q-VD-Oph) abbreviated as QVD (Figure 5b) and evaluated cell death induction in response to **20A** plus chloroquine. As shown in Figure 5b, the induction of cell death was significantly reduced when both HeLa and A549 cells were treated with QV-D-Oph prior to the addition of **20A** plus chloroquine, suggesting the contribution of caspases to cellular demise.

Using a pharmacological approach, we also investigated the possible implication of ferroptosis and necroptosis in cell death induced by **20A** plus chloroquine. As shown in Appendix A, ferrostatin 1, an inhibitor of ferroptosis, markedly inhibited cell death induced by erastin (a known inducer of ferroptosis) but had no significant effect on cellular demise triggered by the combination of **20A** and chloroquine. Similarly, inhibition of necroptosis by necrostatin 1 did not impact cell death induced by the combination of **20A** and chloroquine (Appendix A). Together, these data reveal that neither ferroptosis nor necroptosis are involved in the cytotoxicity effect triggered by **20A** plus chloroquine.

To gain further insight into the mechanism linking LMP and cell death, we evaluated the involvement of two families of cathepsins by using E64d, an inhibitor of the cysteine cathepsin family and pepstatin A, an inhibitor of the aspartic cathepsin family. While E64d and pepstatin A efficiently compromised cell death induced by LLOMe, a well-known inducer of cathepsin-dependent cell death, they failed to inhibit cell death induced by **20A** plus chloroquine (Appendix A). These results suggest that the cysteine and aspartic cathepsins are unlikely to be involved in the execution of cell death induced by **20A** plus chloroquine.

### 2.4. Chloroquine Significantly Activates LMP Triggered by ***20A*** Derivatives

Next, we investigated whether the cooperation between chloroquine and **20A** in induction of cell death can be extended to other triarylpyridine compounds. For this, we tested three new **20A** derivatives (1a, 1b and 1c) (Figure 1) that were synthesized according to the scheme presented in Appendix A. The synthesis and characterization of 1a-c are presented in the experimental section.

We then determined their IC_50_ in HeLa cells before performing a drug combination assay. As shown in Appendix A, these compounds significantly inhibited HeLa cell viability (with IC_50_ values even lower than the one found for **20A**). We then selected sublethal doses (mortality < 10%, as for **20A**) for each compound to analyze LMP and cell death in the presence or absence of chloroquine.

We first analyzed LMP by determining the percentage of cells with Gal3 puncta. As shown in Figure 6a,b, unlike **20A**, all three compounds **1a**–**c** have the ability to induce LMP alone (compared to untreated cells) as evidenced by the percentage of with Gal3 puncta which amounts to 26%, 31% and 25%, respectively. More importantly, the percentage of Gal3 puncta positive cells was further increased when chloroquine was added to each ligand (57%, 56% and 45% for **1a**, **1b** and **1c**, respectively, Figure 6b, left panel). This response was accompanied by an increase in the percentage of cells harboring more than three Gal3 puncta (Figure 6c, right panel), suggesting that a significant population of cells had undergone intensive lysosomal membrane damage. Thus, the combination of chloroquine and **20A** derivatives triggered a massive increase in LMP induction.

### 2.5. Combined Treatment with Chloroquine and ***20A*** Derivatives Significantly Activates Cell Death in Lung Adenocarcinoma Cells

The induction of LMP by **20A**-derivative compounds **1a**, **1b** and **1c** prompted us to investigate if the combination of chloroquine with **20A** derivatives can exacerbate cell death modalities, as seen for **20A**. We thus examined the induction of cell death in response to **20A** and **20A** derivatives in either the presence or absence of chloroquine in A549 lung adenocarcinoma cells. 

As shown in Figure 7a, cell death was markedly enhanced when chloroquine was combined with 1a. Similar results were also observed with **1b** and **1c**, supporting the idea that the triarylpyridine compounds cooperate with chloroquine to trigger a robust cell death. To validate our results in a clinically relevant setting, we also examined the effect of combination of triarylpyridine compounds with chloroquine in patient–derived xenograft cell lines from lung adenocarcinomas (referred as PDX2 and PDX3 in the original paper [29]). As shown in Figure 7b, both PDX cell lines were highly sensitive to the combined treatment.

## 3. Discussion

Targeting lysosome in tumors emerges as an attractive approach for overcoming resistance to therapy. Here, we showed that the triarylpyridine compound **20A** accumulates in the lysosomal compartment and causes lysosome enlargement. The addition of chloroquine to **20A** markedly enhanced the accumulation of enlarged lysosome as compared to a single drug treatment (**20A** or chloroquine alone), supporting the idea that lysosome fission/fusion is affected under these conditions. Calcium fluxes on lysosomal membranes play a major role in fission and fusion processes affecting these organelles. In fact, the lysosomal Ca^2+^ channels P2X4 and Mucolipin 1 (TRPML1) were shown to be involved in lysosomal fusion and fission, respectively [30,31]. The possible contribution of Na^+^ and other cations to lysosomal enlargement has also been suggested in the literature. Whether or not Ca^2+^ and other cations are involved in the lysosomal phenotype observed in response to **20A** and chloroquine/**20A** remains to be clarified.

Another factor that regulates the lysosomal size and number is the TFEB transcription factor, which is responsible for the induction of genes encoding lysosomal proteins through a mechanism that depends on mTOR inhibition [32,33]. The fact that **20A** causes the upregulation of several genes encoding for lysosomal proteins and inhibition of mTOR (as shown in our previous works [24,34] and Appendix A) argues in favor of the activation of the TFEB protein in response to **20A**. It would be interesting to determine if and how TFEB regulates lysosome functions and cell death in cells exposed to **20A** alone or with chloroquine. Further experiments will be necessary to better understand the features of lysosome enlargement induced by **20A** either as a single agent or in combination with chloroquine.

Most importantly, we found that the combination of **20A** and chloroquine promotes a significant induction of lysosomal membrane permeability. Lysosomal membrane permeability can be initiated by several agents namely; (i) lysosomotropic agents, such as chloroquine, which destabilize the lysosomal membrane through either hydrophobic or electrostatic interactions [26,35]; (ii) agents that affect the lipid composition (i.e., cholesterol or sphingolipid) of the lysosomal membrane; (iii) ROS-generating agents that cause lipid peroxidation of this membrane. We found that the addition of antioxidants did not significantly affect LMP triggered by **20A** plus chloroquine, suggesting the implication of ROS-independent mechanisms in this process (Appendix A). Whether or not the accumulation of enlarged lysosomes and/or changes in lysosomal lipid composition are responsible for the induction of LMP warrants further investigation.

Interestingly, we observed that the cooperative effect of chloroquine and **20A** on LMP can be extended to three other triarylpyridine derivatives, suggesting a general mechanism triggered by the triarylpyridine family of compounds. As the induction of LMP is considered an attractive strategy for cancer therapy [14], the characterization of the chemical properties of these molecules and their precise mode of action in regulating LMP deserves further investigation.

Of particular interest are the results showing that, in addition to the robust activation of LMP, the combination of chloroquine and triarylpyridine compounds triggers massive cell death in a variety of cancer cells, including HeLa, U2OS and A549 cell lines, as well as PDX cells from lung cancer patients. Moreover, we found that apoptosis, but neither necroptosis, ferroptosis nor cathepsin-dependent cell death are involved in the execution of cell death, as evidenced by specific pharmacological inhibitors of each pathway. Strikingly, a pan caspase inhibitor only partially recovered cell death triggered by **20A** plus chloroquine, suggesting that the induction of cell death required caspase-dependent and -independent mechanisms. However, the nature of this caspase inhibitor-insensitive component of cell death remains yet unknown. As mitochondrial outer membrane permeabilization (MOMP) is a critical step of lysosome-induced apoptosis [36], it would be interesting to explore if BCl-2 family members (which control MOMP) are implicated in the regulation of the cell death process induced by **20A** plus chloroquine. Of note, caspases seem to operate downstream of LMP (Appendix A) as a pan inhibitor of caspases had no effect on LMP induction. Clearly, further research will be necessary to fully understand the network involved in cell death induced by triarylpyridine compounds either alone or in combination with chloroquine.

By screening of a series of triarylpyridine coumpounds, we first characterized **20A** as a potent G-quadruplex stabilizing DNA Ligand (G4L) [25]. Here, we showed that three new **20A**- derivatives that induce LMP are also able to stabilize an intramolecular G4 structure. However, it remains unclear if the effects of these compounds on lysosomes are related to their interaction with G4 or rather linked to their ability to be sequestrated within the lysosomes. The accumulation of G4Ls in lysosomes has been described only for two other ligands, BMVC and DAOTA [37,38]. Of note, many G4Ls, whose localization could be examined by virtue of their fluorescent property, have shown to display perinuclear cytoplasmic localization. This cytoplasmic localization of G4L was suggested to be attributed to their ability to target mitochondria DNA or cytoplasmic RNAs [39,40]. However, we cannot rule out that some G4Ls are sequestered in lysosomes and that this might affect their anti-proliferative efficacy. In this line, BMVC was shown to be enriched in the lysosomes of cancer cells resistant to chemotherapy and the induction of LMP with LLOMe promoted its nuclear relocation [37]. Overall, it would be interesting to investigate whether different G4 ligand classes affect the lysosome compartment; results may depend on a variety of factors, such as net charge (not all G4L are cationic), hydrophobicity, or potential accumulation in other organelles such as mitochondria.

## 4. Materials and Methods

### 4.1. Cell Culture

The human cervical cancer cell lines HeLa and U20S were purchased from the American Type Culture Collection (ATCC, Manassas, VA, USA) and the human lung carcinoma A549 cell line is a generous gift of Prof. Jean-François Riou (CNRS UMR 7196, Paris, France). The autophagy-deficient (ATG5 and ATG7 depleted) HeLa cell lines were generated as previously described [24]. U2OS expressing mCherry-Galectin3 cell line is a generous gift from Dr. A H. Woldrich (CNRS UMR5, Bordeaux, France) [41]. The lung adenocarcinoma PDX cell lines (PDX2) and (PDX3) are generous gifts from Dr. David Santamaria, INSERM U1218, Bordeaux, France [29]. HeLa cells were grown in RPMI 1640 culture media supplemented with 2 mM glutamine (Invitrogen ,Waltham, MA, USA), 10% fetal bovine serum, 100 units/mL penicillin, and 100 µg/mL streptomycin, while A549, U2OS and PDX cell lines were grown in DMEM supplemented with 10% fetal bovine serum, 100 units/mL penicillin, and 100 µg/mL streptomycin. For mCherry-Galectin3 expressing U2OS cells, 200 µg/mL hygromycin B (#10687010, Invitrogen, Waltham, MA, USA) was added in cell culture media but removed from media before each experiment. All cell lines used in this study were cultivated at 37 °C in a humidified atmosphere with 5% CO_2_.

### 4.2. Reagents and Antibodies

Antibodies against the following proteins were used: ATG5 (#0262-100/ATG5-7C6) from Nanotools (Teningen, Baden-Württemberg, DE), ATG7 (#8558) ), Cleaved-Caspase3 (#9664), LAMP-1 (#9091) from Cell Signaling Technology (Danvers, MA, USA), PARP-1 (C-2-10) (#BML-SA249-0050) from Enzo Life Sciences (Farmingdale, NY, USA), Actin β (#NB600-501) from Novus Biologicals (Centennial, CO, USA), HRP conjugated rabbit (#111-035-003) and HRP conjugated mouse (#115-035-174) from Jackson ImmunoResearch (Ely, Cambridgeshire, UK), Alexa488 conjugated anti-rabbit (#A11008) from Invitrogen (Waltham, MA, USA). Hoechst 33258 (#14530), E64d (#E8640), Propidium iodide (#P4864), Chloroquine diphosphate salt (#C6628), QV-D-Oph (#SML0063), butylhydroxyanisole (BHA, #B1212000), N-acetyl-L-cysteine (NAC, #A8199) were purchased from Sigma-Aldrich (St. Louis, MO, USA).). Tetramethylrhodamine methyl-ester, TMRM (T-668) was purchased from Molecular Probes (Eugene, OR, USA). Fluoromount G (#00-4958-02) was purchased from Invitrogen. **20A** was synthesized as previously described [42,43]. Ferrostatin-1 (#341494) was purchased from Calbiochem (San Diego, CA, USA)), while LLOME (#16008) and Erastin (#17754) were purchased from Cayman Chemical (Ann Arbor, MI, USA).

### 4.3. Synthesis of Bis-Triazole Triarylpyridines

A series of novel bis-triazole substituted 2,4,6-triarylpyridine derivatives **1a**–**c** have been synthesized through a three steps procedure (Appendix A). The synthetic route involves the base-catalysed condensation of commercially available *p*-aminoacetophenone with 4-(methylthio)-benzaldehyde in PEG300 resulting in the formation of the 2,6-bis(4-aminophenyl)-4-[4-(methylthio)phenyl]pyridine (**2**) using the modified Chichibabin pyridine synthesis [25,43,44]. This 2,6-bis(4-aminophenyl)pyridine **2** was then converted into the intermediate corresponding diazido derivative 3 which was then subjected to “click reaction”, a copper-catalyzed azide-alkyne 1,3-dipolar cycloaddition reaction, with three amine-terminal alkynes to afford the substituted 1,2,3-triazole derivatives **1a**, **1b** and **1c**.

#### 4.3.1. Chemicals and Instrumentation for Chemical Synthesis

Commercially available reagents were used as received without additional purification. Melting points were determined with an SM-LUX-POL Leitz hot-stage microscope (Leitz GMBH, Midland, ON, USA) and are uncorrected. IR spectra were recorded on a 380FT-IR spectrophotometer (Nicolet- Thermo Electron Scientific Instruments LLC, Madison, WI, USA). NMR spectra were recorded with tetramethylsilane as an internal standard using an AVANCE 300 spectrometer (Bruker BioSpin, Wissembourg, France). Splitting patterns have been designated as follows: s = singlet; d = doublet; t = triplet; m = multiplet. Analytical TLC were carried out on 0.25 precoated silica gel plates (POLYGRAM SIL G/UV254) and visualization of compounds after UV light irradiation. Silica gel 60 (70–230 mesh) was used for column chromatography. High resolution mass spectra (electrospray in positive mode, ESI+ or MALDI-TOF MS) were recorded on a Q-TOF Ultima apparatus (Bruker Daltonics, Bremen, Germany). Elemental analyses were found within ±0.4% of the theoretical values.

#### 4.3.2. Synthesis of 2,6-bis(4-aminophenyl)-4-[4-(methylthio)phenyl]pyridine (**2**) (Appendix A)

*p-*Aminoacetophenone (5 g, 37 mMol) was added to a suspension of crushed NaOH (1.5 g, 37 mmol) in PEG300 (20 mL) at room temperature. The suspension was stirred and heated to 80 °C to form a clear solution. Then, *p*-methylthiobenzaldehyde (2.8 g, 18.5 mmol) was added to the reaction mixture and heated at 110 °C for 4 h, after which NH_4_OAc (42.7 g, 555 mmol) was added and the heating was reduced to 100 °C and stirred for 4 h. After cooling to room temperature, 400 mL cold water was added to the mixture to get a precipitate which was filtered and further purified through silica gel chromatography using DCM:MeOH (0–5%) as eluent affording compound **2** as a brown-yellow powder (52%); ^1^H-NMR δ (300 MHz, CDCl_3_) 8.37 (d, 4H, *J* = 8.70 Hz, 2H-2′ and 2H-6′), 7.95 (s, 2H, H-3 and H-5), 7.92 (d, 4H, *J* = 8.70 Hz, 2H-3′ and 2H-5′), 7.73 (d, 2H, *J* = 8.70 Hz, H-2″ and H-6′′), 7.44 (d, 2H, *J* = 8.70 Hz, H-3″ and H-5″), 5.89 (br, 4H, 2NH_2_), 2.59 (s, 3H, CH_3_S). MALDI-TOF MS *m/z* [M + H]^+^ Calcd C_24_H_21_N_3_S: 384.1529, Found: 384.1523.

#### 4.3.3. Synthesis of 2,6-bis(4-azidophenyl)-4-[4-(methylthio)phenyl]pyridine (**3**) (Appendix A)

Compound **2** (200 mg, 0.5 mmol) was added to 15 mL 6N HCl and the reaction mixture was cooled to 0 °C and NaNO_2_ (138 mg, 2 mmol) was added and was stirred for 30 min under cold condition. Sodium azide (195 mg, 3 mmol) was added to the reaction mixture portion-wise under cold condition; and the reaction mixture was allowed to stir at room temperature for 2 h. The mixture was neutralized with sodium bicarbonate and extracted with ethyl acetate (3 × 30 mL). The organic part was evaporated under vacuum at 30 °C to afford compound **3** as yellow crystals (98%); as a yellow solid which was used for next reaction. IR (KBr) 2105 cm^-1^ (N_3_). MALDI-TOF MS *m/z* [M+H]^+^ Calcd for C_24_H_18_N_7_S; 436.1339, Found: 436.1326.

#### 4.3.4. General Procedure For The Synthesis of 2,6-bis{4-[4-(substituted-aminoalkyl)-1*H*-1,2,3-triazol-1-yl]phenyl}-4-[4-(methylthio)phenyl]pyridines **1a**–**c**

Compound **3** (50 mg, 0.1 mmol) was added to a mixture of 4 mL H_2_O and THF (1:1), then CuSO_4_·5H_2_O (0.002 mmol) and sodium ascorbate (0.1 mmol) were added to the mixture and stirred. After 5 min, the appropriate substituted alkyne (0.3 mmol) was added and stirred for 12 h at room temperature. The solvent was evaporated under reduced pressure, and the residue was washed with water followed by diethyl ether to afford the desired bis-triazolylarylpyridine compound **1a**–**c**.

*4-[4-(Methylthio)phenyl]-2,6-bis{4-[4-(3-pyrrolidin-1-ylpropyl)-1H-1,2,3-triazol-1 yl]phenyl}pyridine* (**1a**).

Yellow crystals (84%); ^1^H-NMR (300 MHz, CDCl_3_) δ 8.39 (d, 4H, *J* = 8.80 Hz, 2H-2′ and 2H-6′), 7.96 (s, 2H, H-3 and H-5), 7.92 (d, 4H, *J* = 8.80 Hz, 2H-3′ and 2H-5′), 7.86 (s, 2H, 2H triazol.), 7.74 (d, 2H, *J* = 8.40 Hz, H-2″ and H-6″), 7.43 (s, 2H, *J* = 8.40 Hz, H-3″ and H-5″), 2.91 (t, 4H, *J* = 7.50 Hz, 2NCH_2_), 2.63-2.53 (m, 12H, 6 NCH_2_), 2.59 (s, 3H, CH_3_S), 2.06-1.96 (m, 4H, 2CH_2_ pyrrol.), 1.84-1.77 (m, 4H, 2CH_2_ pyrrol.). ^13^C-NMR (75 MHz, CDCl_3_) δ 157.5 (C-2 and C-6), 151.4 (C-4), 150.2 (C-4 triazol.), 142.1 (C-4″), 140.6 (C-1′), 138.9 (C-4′), 135.9 (C-1″), 129.8 (C-3′ and C-5′), 128.7 (C-2″ and C-6″), 127.9 (C-3″ and C-5″), 121.8 (C-2′ and C-6′), 120.2 (C-5 triazol.), 118.5 (C-3 and C-5), 58.3 (NCH_2_), 56.8 (NCH_2_), 28.4 (CH_2_), 27.4 (CH_2_), 24.9 (CH_2_), 16.8 (CH_3_S). MALDI-TOF MS *m/z* [M+H]^+^ Calcd for C_42_H_48_N_9_S; 710.3748, Found: 710.3740.

*2,6-Bis{4-[4-(3-(4-methylpiperazin-1-yl)propyl)-1H-1,2,3-triazol-1-yl]phenyl}-4-[4-(methylthio)phenyl]-pyridine* (**1b**)

Orange crystals (68 %); ^1^H-NMR (300 MHz, CDCl_3_) δ 8.38 (d, 4H, *J* = 8.70 Hz, 2H-2′ and 2H-6′), 7.96 (s, 2H, H-3 and H-5), 7.92 (d, 4H, *J* = 8.70 Hz, 2H-3′ and 2H-5′), 7.86 (s, 2H, 2H triazol.), 7.73 (d, 2H, *J* = 8.50 Hz, H-2″ and H-6″), 7.43 (d, 2H, *J* = 8.50 Hz, H-3″ and H-5″), 2.89 (t, 4H, *J* = 7.50 Hz, 2NCH_2_), 2.66-2.42 (m, 20 H, 2NCH_2_ and 8NCH_2_ piperaz.), 2.59 (s, 3H, CH_3_S), 2.32 (s, 6H, 2NCH_3_), 2.02-1.92 (m, 4H, 2CH_2_). ^13^C-NMR (75 MHz, CDCl_3_) δ 157.5 (C-2 and C-6), 151.3 (C-4), 150.1 (C-4 triazol.), 142.1 (C-4″), 140.6 (C-1′), 139.0 (C-4′), 136.1 (C-1″), 129.7 (C-3′ and C-5′), 128.8 (C-2″ and C-6″), 128.0 (C-3″ and C-5″), 121.7 (C-2′ and C-6′), 120.2 (C-5 triazol.), 118.4 (C-3 and C-5), 59.1 (NCH_2_), 56.5 (NCH_2_), 54.5 (NCH_2_), 47.4 (NCH_3_), 28.0 (CH_2_), 25.0 (CH_2_), 16.8 (CH_3_S). MALDI-TOF MS *m/z* [M+H]^+^ Calcd for C_44_H_54_N_11_S; 768.4279, Found: 768.4269.

*2,6-Bis{4-[4-(2-(4-methylpiperazin-1-yl)ethyl)-1H-1,2,3-triazol-1-yl]phenyl}-4-[4-(methylthio)phenyl]-pyridine* (**1c**).

Yellow crystals (64%); ^1^H NMR (300 MHz, CDCl_3_) δ 8.39 (d, 4H, *J* = 8.80 Hz, 2H-2′ and 2H-6′), 7.96 (s, 2H, H-3 and H-5), 7.94 (s, 2H, 2H triazol.), 7.91 (d, 4H, *J* = 8.80 Hz, 2H-3′ and 2H-5′), 7.73 (d, 2H, *J* = 8.40 Hz, H-2″ and H-6″), 7.43 (d, 2H, *J* = 8.40 Hz, H-3″ and H-5″), 3.07 (t, 4H, *J* = 7.80 Hz, 2NCH_2_), 2.81 (t, 4H, *J* = 7.80 Hz, 2NCH_2_), 2.63-2.43 (m, 16 H, 8NCH_2_ piperaz.), 2.59 (s, 3H, CH_3_S), 2.34 (s, 6H, 2NCH_3_). ^13^C-NMR (75 MHz, CDCl_3_) δ 157.5 (C-2 and C-6), 151.5 (C-4), 148.0 (C-4 triazol.), 142.2 (C-4″), 140.8 (C-1′), 139.0 (C-4′), 136.2 (C-1″), 129.8 (C-3′ and C-5′), 128.8 (C-2″ and C-6″), 128.0 (C-3″ and C-5″), 121.9 (C-2′ and C-6′), 120.8 (C-5 triazol.), 118.5 (C-3 and C-5), 58.6 (NCH_2_), 55.8 (NCH_2_), 53.1 (NCH_2_), 46.5 (NCH_3_), 24.6 (CH_2_), 16.8 (CH_3_S). MALDI-TOF MS *m/z* [M+H]^+^ Calcd for C_42_H_50_N_11_S; 740.3966, Found: 740.3924.

### 4.4. Quadruplex Stabilization

Binding of **1a**, **1b** and **1c** to G-quadruplexes was investigated by FRET melting analysis as previously described [45]. Compounds were tested at 1–10 µM concentration in the presence of the F21T oligonucleotide (FAM-5′-GGGTTAGGGTTAGGGTTAGGG-3′-TAMRA) at 0.2 µM strand concentration in a 10 mM lithium cacodylate pH 7.2 buffer supplemented with 10 mM KCl and 90 mM LiCl.

### 4.5. Mitochondrial and Lysosomal Staining and Evaluation of ***20A*** Localization

Cells were grown on glass coverslips on a 6-well plate (6 × 10^4^ cells/well), treated with or without 5 µM **20A** for 6h. Cells were stained with 50 nM MitoTracker Green FM (#M7514, Molecular Probes, Eugene, OR, USA) and 50 nM LysoTracker^TM^ Red DND-99 (#L7528, Molecular Probes) 30 min before the end of the incubation time of **20A**. Cells were then placed in a perfusion chamber with DMEM without red Phenol containing 15 nM Lysotracker Red DN-99. Image acquisition was made on an LSM 510 META confocal microscope Zeiss (Oberkochen, Ostalbkreis, DE) equipped with an Apoplan x63 objective. Identical exposures were used for each channel throughout individual experiments, and no images were altered after capture. Co-localization of blue signal (**20A**) with red signal (lysosome) and green signal (mitochondria) was analyzed by superposing the fluorescence profile of each signal. For some experiments, lysosome size was scored using “the analysis particles” function of Image J after applying the autoThreshold “Minimum”. For the analysis, we took into consideration particles that display a size between 0.1 and 1.6 µm^2^ and circularity between 0.5 and 1 µm.

### 4.6. Lysosomal Membrane Permeabilization (LMP) Analysis

#### 4.6.1. Quantification of Galectin 3 Puncta

Lysosomal membrane damage was assessed thanks to U2OS expressing mCherry-Galectin3 cells. Cells were plated on 12-well plates (7.5 × 10^4^ cells/well) treated with or without **20A** in either the presence or absence of 25 µM chloroquine for 24 h. Cells were then washed with PBS, fixed with 4% (v/v) paraformaldehyde for 10 min at room temperature and nuclei were then counterstained for 15 min with Hoechst 33258. Image acquisition was made with a DMI8 epifluorescence microscope (Leica Microsystems, Wetzlar, Hessen, DE) with a x20 objective (NA 0.40) equipped with a digital CMOS camera (Hamamatsu Photonics, Hamamatsu, Shizuoka, JP) and filter bloc for detection of phase contrast and blue (exc: 325–375 nm/em: 435–485 nm) and red (exc: 541–551 nm/em: 565–605 nm) fluorescence. Each image was collected with identical exposure times and scaled equally. Five images were acquired for each condition and then analysed thanks to Image J software to ImageJ software (NIH-National Institutes of Health, Bethesda, MD, USA).The number of Galectin3 puncta within each cell was scored using the “Find Maxima” tool.

#### 4.6.2. Co-Localization of Galectin 3 with LAMP1

To assess lysosomal membrane permeabilization (LMP), co-localization study of Galectin3 puncta and LAMP1 was carried out by immunofluorescence analysis. Cells were grown on glass coverslips, fixed in ice-cold methanol at −20°C for 15 min and then blocked/permeabilized with 3% BSA containing 0.3% Triton X-100 for 1 h. Cells were then incubated with the primary antibody overnight at 4°C. After washing, cells were incubated with the secondary antibody (anti-rabbit Alexa Fluor 488 antibody) for 1 h at room temperature in a dark chamber and stained with 2 μg/mL of Hoechst 33258 (H3569, Promega, Madison, WI, USA) for 10 min. After two PBS wash, cells were mounted with Fluoromount (F4680, Sigma-Aldrich, St. Louis, MO, USA) and images were acquired using a Zeiss LSM510 Meta confocal microscope Zeiss, (Oberkochen, Ostalbkreis, DE) equipped with an oil immersion Apoplan 63× objective (NA 0.75). *Z*-stack series were made with 0.6 µm interval and the corresponding images represent a *z*-projection of maximal intensity. Identical exposures were used for each channel throughout individual experiment. No images were altered after capture. Processing of images was performed with the ImageJ software.

### 4.7. Evaluation of Cell Death

Cell death was determined by measuring the plasma membrane permeability using propidium iodide dye (P4864, Sigma). Briefly, supernatant and attached cells were collected, pelleted at 350 g for 5 min and loaded with 1 µg/mL Propidum iodide for 15 min. Cells were then analyzed by flow cytometry (BD FACSCalibur, Becton Dickinson, Franklin Lakes, NJ, USA) in the FL2 channel (exc: 488 nm/em: 564–606 nm). For some experiments, cells were stained with 150 nM tetramethylrhodamine methyl ester (TMRM, from Molecular Probes-Life Technologies) and 5 µg/mL 4′,6-diaminidino-2-phenylindole (DAPI, from Molecular Probes-Life Technologies) for 30 min at 37 °C. Cytofluorometric determinations of early and late apoptotis were carried out on a MACSQuant cytometer (Miltenyi Biotec, Bergisch Gladbach, Germany) upon gating on events displaying normal forward scatter and side scatter parameters. Apoptosis was also evaluated by western blotting analyses of cleaved forms of either PARP 1 or caspase 3 [24]. Where indicated, the caspase inhibitor QV-D-Oph (20 µM) was added to cells to evaluate the possible implication of caspases in cell demise.

### 4.8. Immunoblot Assay

Cell extracts were prepared in 10 mM Tris, pH 7.4, 1% SDS, 1 mM sodium vanadate, 2 mM PMSF (93482, Sigma-Aldrich), 1% Protease Inhibitor Cocktail (P8849, Sigma-Aldrich), and 1% Halt Phosphatase Inhibitor Cocktail (1862495, Thermo Fisher Scientific, Waltham, MA, USA) and treated with benzonase endonuclease (71205, Merck Millipore) for 5 min at room temperature, and boiled for 5 min. Fractions (30–50 µg) of cellular extract proteins were subjected to SDS-PAGE as previously described [21]. The densitometry quantification was performed using the ImageJ software.

### 4.9. Cell viability Assay

MTT assay was used to monitor cell viability in cells subjected to different treatments. Briefly, cells were seeded in a 96-well plate (4 × 10^3^ cells/well). After 24 h of treatment of cells with drug, MTT was added to each well to a final concentration of 0.5 mg/mL during 3 h at 37 °C. The supernatant of cells was then removed, and 100 µL of DMSO was added per well. The absorbance in each well was measured at 570 nm and 630 nm using a Flexstation 3 microplate reader FlexStation 3 Microplate Reader (Molecular Devices, San Jose, CA, USA). To determine the IC50 of **20A** derivative compounds, a non-linear regression curve was fit using GraphPad Prism software (GraphPad Software Inc., San Diego, CA, USA).

### 4.10. Statistical Analysis

Unless otherwise stated, each experiment was performed three times. Results were expressed as the mean value ± standard deviation (SD). Statistical analysis was performed by Mann-Withney unpaired statistical test. *p* < 0.05 was considered statistically significant. Where indicated, Kruskal-Wallis test, with post-hoc Dunn’s analysis was used for statistical analysis. The software used was GraphPad Prism 5.

## 5. Conclusions

In conclusion, our study uncovers the lysosomal effects of G4 ligands belonging to the triarylpyridine family and proposes a rationale for combining these compounds with chloroquine to increase their effectiveness against cancer. It would be interesting to carry out a comprehensive study on multiple G4 ligands to classify them according to their lysosomal sequestration and their capacity to cooperated with chloroquine-like agents. In fact, the chemical properties of the molecules including pKa value(s) and lipophilicity are critical factors that determine the ability of the molecule to be trapped within the lysosomes and provoke LMP [46]. This information should then be taken into account for the development of new G-quadruplex ligands designed for optimal cancer therapy.

## Figures and Tables

**Figure 1 cancers-12-01621-f001:**
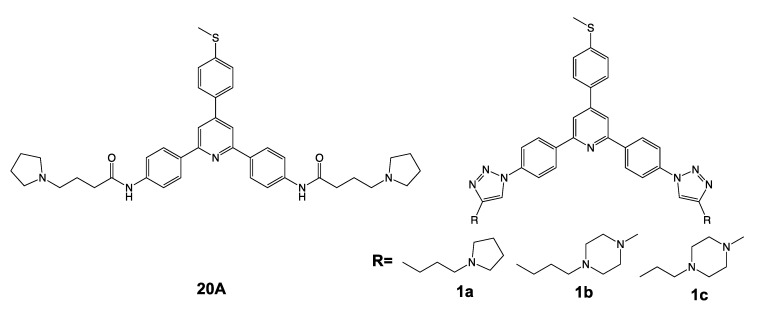
Formula of triarylpyridine compounds used in this study.

**Figure 2 cancers-12-01621-f002:**
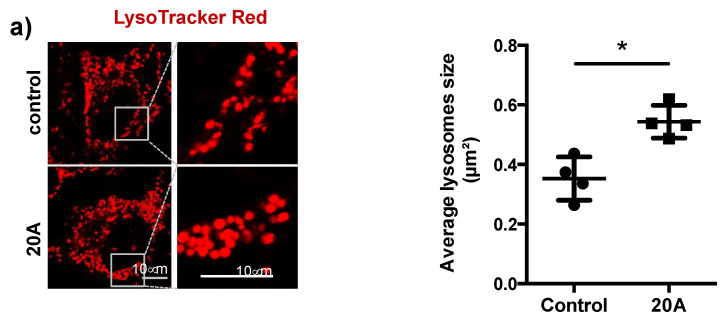
**20A** accumulates within the lysosomes and causes lysosomal enlargement. (**a**) HeLa cells were treated with or without 5 µM **20A** for 24 h and then stained with LysoTracker Red (red signal) to visualize lysosomes. Left panel, representative confocal fluorescence images are presented for each condition. Right panel, the average of lysosome size was scored for the experiment in (**a**) using Image J tools (*n* > 30 cells/condition, *p* value * < 0.05 using Mann-Whitney test). Scale bars: 10 µm. (**b**) HeLa cells were treated with or without 5 µM **20A** for 6 h and then stained with LysoTracker Red and MitoTracker Green to visualize lysosomes and mitochondria, respectively. Representative confocal images are presented for LysoTracker Red (red signal), **20A** fluorescence (blue signal) and MitoTracker Green (green signal). Magnified views of boxes are presented for individual or merged signals. Scale bars: 25 µm. (**c**) The fluorescence intensities of red, blue and green signals plotted along the yellow bar are shown in the “**20A**-treated” cells panel.

**Figure 3 cancers-12-01621-f003:**
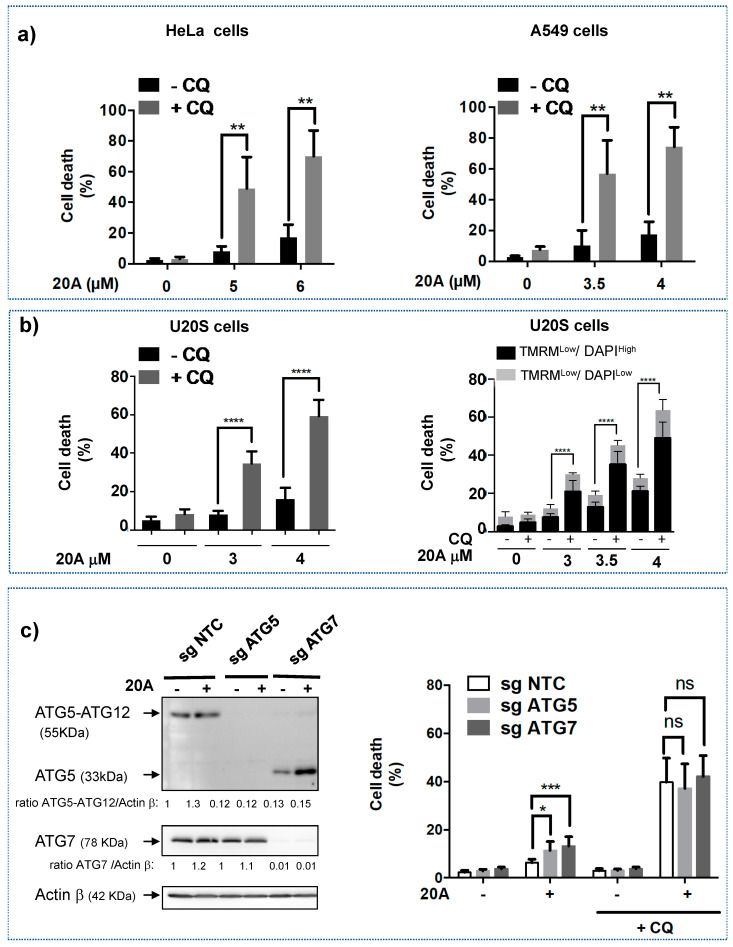
**20A** and chloroquine act in concert to trigger cell death. (**a**) HeLa cells (left) and A549 cells (right) were treated with the indicated concentration of **20A** with or without 25 µM of chloroquine for 24 h. Cell death was evaluated by scoring the percentage of propidium iodide (PI) positive cells after flow cytometer analysis. The data represents the mean ± SD of 6 values obtained from three independent experiments each performed in duplicate (** *p* value < 0.01 using Kruskal-Wallis test, with post-hoc Dunn’s analysis). (**b**) U20S cells were treated with the indicated concentration of **20A** with or without 25 µM of chloroquine for 16 h. Left panel, cell death was evaluated by scoring the percentage of propidium iodide positive cells after flow cytometer analysis. The data represents the mean ± SD of 9 values obtained from three independent experiments each performed in triplicate (*p* value **** < 0.0001, using Mann-Whitney test). Right panel*,* Cell death was evaluated after 16 h treatment with **20A** ± chloroquine by scoring the percentage of TMRM^Low^/ DAPI^Low^ (early apoptosis) and TMRM^Low^/ DAPI^High^ (late apoptosis) cells following flow cytometer analysis. The data represents the mean ± SD of 9 values obtained from three independent experiments each performed in triplicate (*p* value **** < 0.0001, using Mann-Whitney test). (**c**) Autophagy-proficient (sgNTC) and -deficient (sgATG5 and sgATG7) HeLa cells were treated with 5 µM **20A** either in the presence (25 µM) or absence of chloroquine for 24 h. Left panel, silencing of key autophagy proteins, ATG5 and ATG7 was checked by western blotting. The western blots were quantified and given as ratio of ATG5-ATG12/Actin β or ATG7/Actin β. Right panel, cell death was evaluated by scoring the percentage of propidium iodide positive cells after flow cytometer analysis. The data represents the mean ± SD of nine values obtained from three independent experiments each performed in triplicate (*p* value * < 0.05 and *p* value *** < 0.001, ns: not significant using Mann-Whitney test).

**Figure 4 cancers-12-01621-f004:**
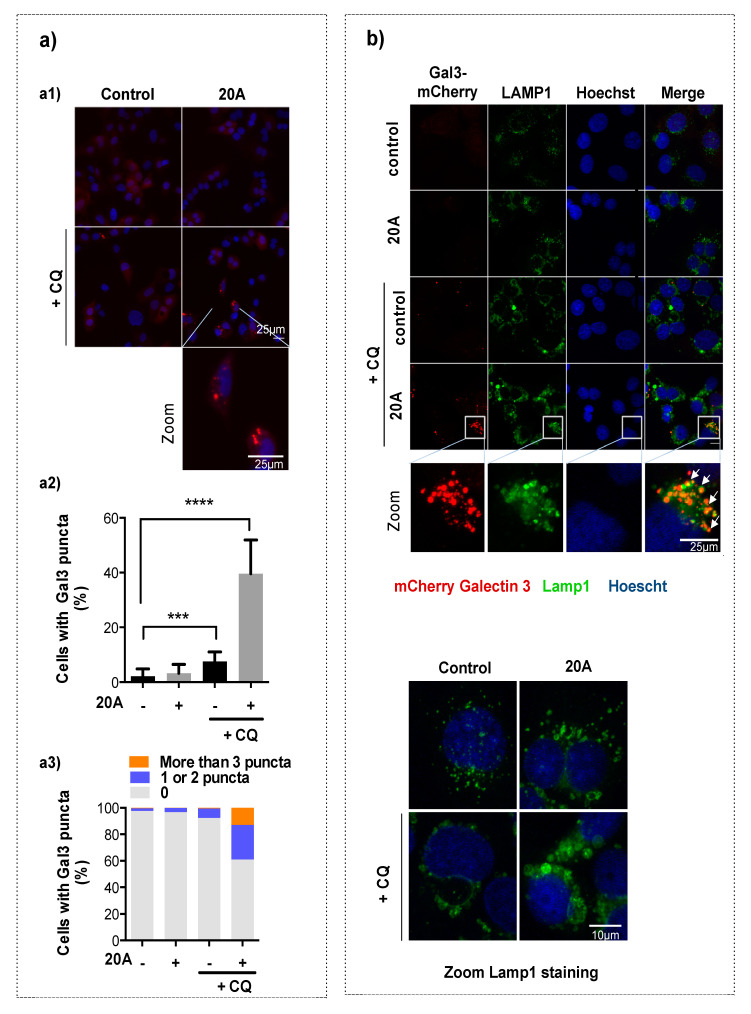
**20A** and chloroquine act in concert to trigger LMP in cancer cells. (**a**) mCherry-Galectin3 expressing U2OS cells were treated or not with 3 µM **20A** in either the presence or absence of 25 µM chloroquine for 24 h. (**a1**) representative fluorescence images of Galectin 3 staining (red signal) and nuclei stained with Hoechst (blue signal) are shown; scale bar represents a distance of 25 µm. (**a2**) The percentage of cells displaying at least one Galectin 3 punctae is scored. Data are presented as mean ± SD of 15 values obtained from 5 randomly chosen fields in each of the three independent experiments. Data are presented as the mean percentage of at least 1000 cells analyzed from five randomly chosen fields in each of the three independent experiments (*** *p* value < 0.001 **** and *p* value < 0.0001 using Kruskal-Wallis test, with post-hoc Dunn’s analysis). (**a3**) The percentage of cells with >3, 1 and 2, 0 Galectin 3 puncta are scored. (**b**) mCherry-Galectin3 U2OS cells were treated or not with 3 µM **20A** either in the presence or absence of 25 µM chloroquine for 24 h and then immunostained with an antibody that recognizes LAMP1, a lysosomal marker. Nuclei were counterstained with Hoechst. Representative of (z-projected) confocal images of Galectin 3 (red), nuclei (blue) and LAMP1 (green) are shown. Scale bar represents a distance of 25 µm. Zoom image of **20A**-treated cells shows that most Galectin 3 puncta co-localize with LAMP1 (representative co-localization signals are depicted by white arrowheads). Zoom Lamp1 staining images of cells treated with **20A** ± chloroquine are shown. Scale bar: 10 µm.

**Figure 5 cancers-12-01621-f005:**
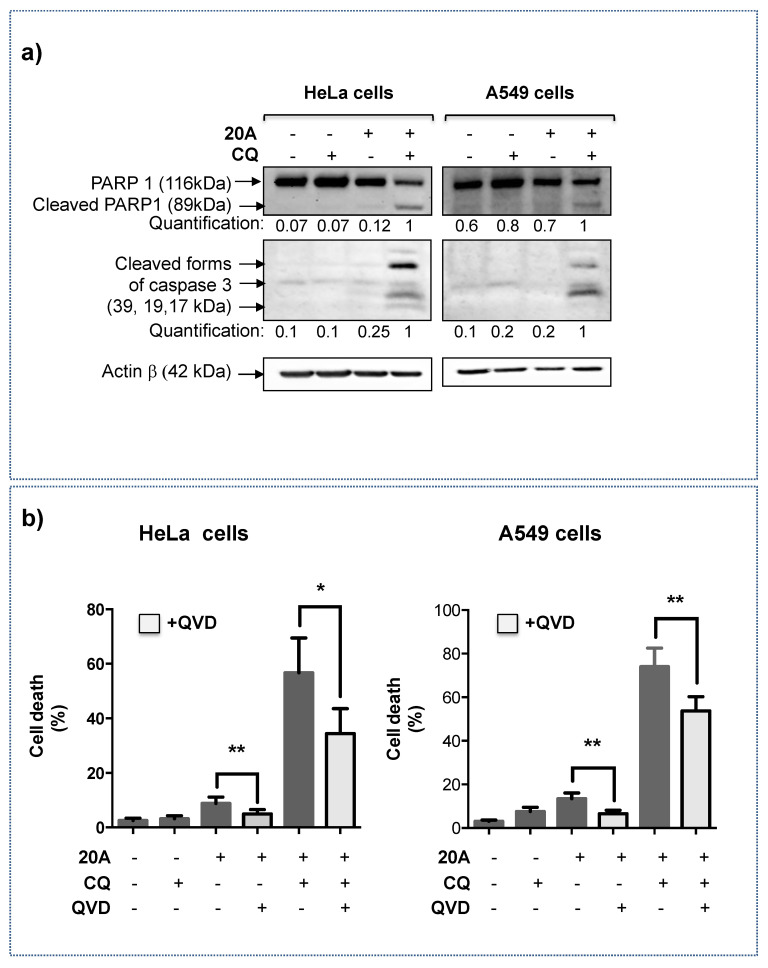
Apoptosis is involved in cell death induced by **20A** plus chloroquine. (**a**) HeLa cells (left panel) and A549 (right panel) cells were treated with 5 µM and 3.5 µM **20A**, respectively, in the presence or absence of 25 µM chloroquine for 24 h. Apoptotic cell death was assessed by immunoblot analysis of cleaved forms of either Caspase 3 or PARP1. Actin β was detected as a loading control. The Western blots were quantified and given as ratio of cleaved caspase3/Actin β or cleaved Parp1/Actin β (**b**) HeLa cells (left panel) and A549 (right panel) cells were treated with or without 20 µM QV-D-Oph two hours prior to the addition of **20A** (5 µM and 3.5 µM for HeLa cells and A549 cells, respectively). Where indicated, cells are also exposed to 25 µM chloroquine (CQ) for 24 h. Cell death was assessed by scoring propidium iodide positive cells. The data represents the mean ± SD of six values obtained from three independent experiments each performed in duplicate (* *p* value < 0.05 and ** *p* value < 0.01 using Mann-Whitney test).

**Figure 6 cancers-12-01621-f006:**
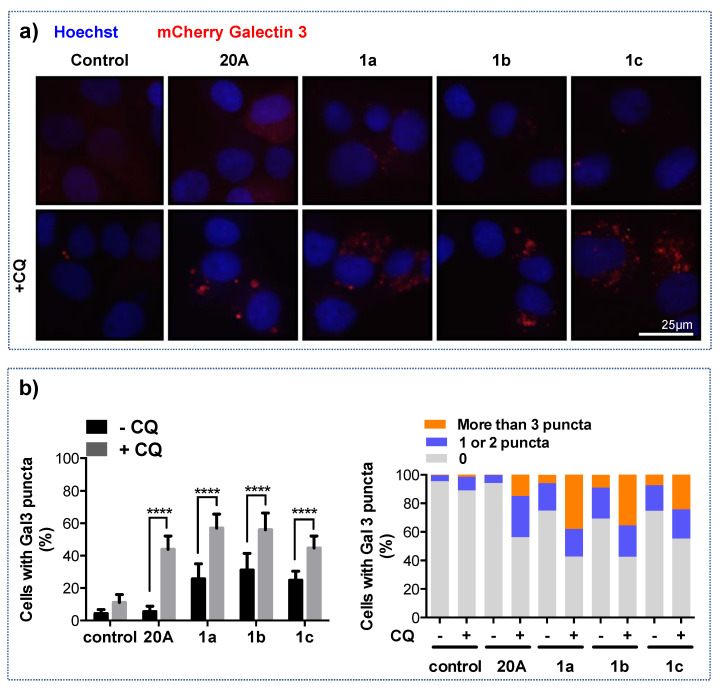
Chloroquine enhances LMP triggered by **20A** derivatives. (**a**) mCherry-Galectin3 expressing U2OS cells were treated with or without 3 µM **20A**, 2 µM **1a**, 1 µM **1b** or 1 µM **1c** in either the presence or absence of 25 µM chloroquine for 24 h. Representative fluorescence merge images are presented with Galectin 3 (red signal) and Hoechst (blue signal), scale bar 25 µm. (**b**) Left Panel, quantification of the percentage of cells displaying at least one Galectin3 puncta. Data are presented as mean ± SD of 15 values obtained from five randomly chosen areas in each of the three independent experiments. Right panel, the percentage of cells with 0, 1 or 2, or 3 or more Galectin 3 puncta are scored. Data are presented as the mean percentage of at least 1000 cells analyzed from 5 randomly chosen field in each of the three independent experiments (**** *p* value < 0.0001 using Mann-Whitney test).

**Figure 7 cancers-12-01621-f007:**
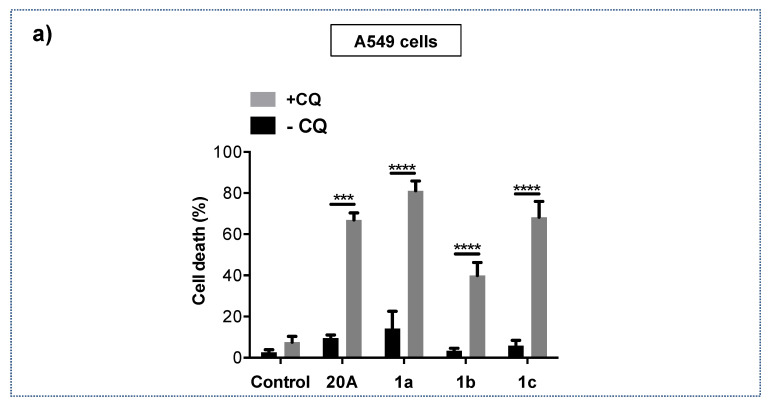
Combination of chloroquine with **20A** derivatives significantly enhances cell death in both A549 lung cancer cells and patient–derived xenograft cell lines from lung cancer. A549 lung cancer cell lines (**a**) and two PDX from lung cancer PDX2 and PDX3 (**b**) were treated with either 3.5 µM **20A**, 2.5 µM 1a, 1.5 µM 1b or 2 µM 1c in the presence and absence of 25 µM chloroquine for 24 h. Cell death was evaluated by scoring the percentage of propidium iodide positive cells by flow cytometer analysis. The data represents the mean ± SD of 9 values obtained from three independent experiments each performed in triplicate (*p-*value *** < 0.001 and **** < 0.0001 using Mann-Whitney test).

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
