# Peer review of "Triarylpyridine Compounds and Chloroquine Act in Concert to Trigger Lysosomal Membrane Permeabilization and Cell Death in Cancer Cells"

_cancers, 2020, doi:10.3390/cancers12061621_

Round 1

Reviewer 1 Report

The manuscript submitted by Mojgan Djavaheri-Mergny and collaborators deal with the effect produced on the lysosomes of cancer cells by G-Quadruplex ligands belonging to the triarylpyridine family, especially when in combination with cloroquine.

The paper is nicely written and the experimental part include an impressive number of assays. The data presented are basically a continuation of a previously published work (Nucleic Acids Research, 2019, 47, 2739). Nevertheless, it gives a good number of new information about the cellular behavior of this class of compounds.

Having said that, it remains unclear to me what is the correlation between the fact that these drugs are G4 ligands and their effect on lysosomes. No data in this manuscript correlate the G4-stabilizig ability of this class of molecules with their activity when in combination with cloroquine. There is not a single experiment with other G4 ligands not belonging to the triarylpyridine family. Besides, there is plenty of non-G4 ligands which can be sequestrated by lysosomes, so the described effects can be not related to any G4 interaction. As a consequence, all the discussion and conclusions with general sentences concerning G4 ligands and lysosomes are, in my opinion, misplaced and should be removed. Even the title is too general and could lead to misinterpretations! Indeed, there is no proof that “G-quadruplex ligands and chloroquine act in concert to trigger lysosomal membrane permeabilization […]”. Yes, 20A (and other triarylpyridines) does it, but, as said, it could be that this effect has nothing to do with the G4 binding ability.

The authors themselves state “It would be interesting to carry out a comprehensive study on multiple G4Ls to classify them according to their lysosomal sequestration and their capacity to cooperated with chloroquine-like agents”, so why not use at least one of them in this paper to have at least a hint?

At this stage, the presented data, give (precious) information on molecules belonging to the triarylpyridine family and should be discussed like that.

So I will say it more clearly. I like the paper and its science but in my opinion there are two options in order to publish these data: 1) The authors perform control experiments with other G4 ligands not belonging to the triarylpyridine family. 2) The authors re-write the parts of the manuscript where there is a tendency to generalize on the effect of G4 ligands on lysosomes when in combination with chloroquine (even the title).

Minor point:

The authors state “. In vitro fluorescence experiments confirmed that 20A emits a weak blue fluorescence following excitation in the near UV spectrum (data not shown).” Why not show this? I would like to see it.

All the best

Author Response

Please see in the attached document a point-by-point response to the reviewer 1.  

Reviewer 2 Report

In this research Beauvarlet and colleagues investigate the relationship between the lysosomal damage triggered by a number of compounds, 20A and its derivatives belonging to the triarylpyridine family. The authors show that 20A accumulates inside lysosomes and  markedly enlarges them. Addition of chloroquine (CQ) triggers lysosomal membrane permeabilization (LMP) and cell death. The authors then detailedly investigate the putative death mechanisms by probing different pathways in several cellular models. Of interest, they eventually report that CQ treatment significantly increases cell death triggered by 20A and derivatives in patient-derived xenograft cell lines established from lung carcinomas. They conclude that CQ may prove useful to enhance the anticancer activity of these compounds.

The subject of the manuscript focuses on the very interesting and so far largely underlooked phenomenon of LMP and its basic contribution to the therapeutic activity of several anticancer agents. The paper is well written, the methods are at most appropriate and the conclusions are supported by the results presented.

In my opinion, the manuscript has some still obscure point(s) that need clarification. In particular, the death mechanisms have been investigated in detail but, in conclusion, that (or those) responsible of cell death does not unambiguously emerge from the data and the author conclusions. In addition to this general aspect, I have identified some critical points that are listed below for the author consideration.

Main criticisms

Page 5, line 126. Chloroquine (CQ) is known to attenuate lysosomal trapping of lysosomotropic drugs. Have the authors checked whether CQ reduces the accumulation of 20A in the lysosomal compartment? Close to this point, is 20A fluorescence pH-dependent?

Page 5, line 140. What is the rationale for using Galectin 3-mCherry-overexpressing cells (as stated in Fig. 2b legend) for this experiment? The galectin signal was not shown or considered in this panel.

Page 7, line 176. LMP is convincingly shown to occur in a significant fraction of cells treated with CQ+20A. Since damaged lysosomes range from 1 to 3/cell, what is the percentage of damaged lysosomes/cell?

The authors conclude here that there is lethal lysosomal permeabilization, but subsequently they state that lysosomal cysteine and aspartic cathepsins are not involved in death (page 9, lines235-238). This, likely due to the quite low level of LMP triggered by the treatments, seems to be contradictory with respect to the stated lethality of LMP. Have the authors checked or have any information on whether the cathepsins are released from leaky lysosomes?

Pag 9, line 214-218. Cleavage of PARP and caspase 3 are reliable markers of early apoptotic death when they precede the loss of plasma membrane integrity, a good marker of late apoptosis when showing up later than the early markers. Since all these assays (PARP and caspase cleavage, PI uptake) were performed at the same experimental time (24 h of treatment with single or combined drugs), this makes unclear whether appearance of the early markers precedes the late ones. In addition, cell viability was essentially measured by PI uptake (see also Fig. 2, 3 and so on), which essentially reveals accidental, or other non-canonically apoptotic or necrotic forms of death. The authors should make a time-course analysis, possibly using a test (such as the Annexin V/PI, or any other similar), clearly showing occurrence and timing of early and late apoptosis.

Page 9, lines 220-224. The data clearly show that cell death is only partly caspase-dependent (about 50% in HeLa cells and a bit less in A549 ones). Could  the authors comment on the possible nature of the caspase inhibitor-insensitive component?

Page 12, line 287. Is the death-inducing capability of 1a significantly different or not from that of 20A? This is not clearly stated in the text.

Minor criticisms

Page 18, line 526. Using ‘1800 rpm for 5 min’ requires the specification of the rotor type used for centrifugation. To increase accuracy and reproducibility of the experimental conditions, RCF or g•min should be indicated.

Author Response

Please see in  the attached document a point-by-point response to the reviewer 2.

Reviewer 3 Report

The paper submitted by Djavaheri-Mergny and coworkers report on the effect of 20A on lysosome and how the combination with chloroquine led to a significant induction of lysosomal membrane permeabilization coupled to massive cell death.

The manuscript is well written and organized appropriately; the large amount of experiments is consistent with the conclusions suggested by the authors.

Minor point:

  • I suggest to add some references regarding G4 ligands to the introduction (line 86), for examples: Am. Chem. Soc., 2007, 129 (6), 1502-1503 DOI: 10.1021/ja065591t; Nucleic Acids Research, 2012, Vol. 40, No. 12 doi:10.1093/nar/gks152; Phys.Chem.Chem.Phys., 2017, 19, 17404 DOI: 10.1039/c7cp02816d
  • The structure of 20A showed in figure 5a should be reported at the end of introduction in a new figure
  • The authors define 20A as G4 ligand belonging to the triarylpyridine family (as demonstrated previously). However in this manuscript this information is not so relevant to appear in the title. The authors did not demonstrate that the effect of 20A on the lysosome is due to the interaction with G4. Therefore I suggest to replace the "G-quadruplex ligands..." in the title with "triarylpyridine derivative...."

Author Response

Reviewer #3:

The paper submitted by Djavaheri-Mergny and coworkers report on the effect of 20A on lysosome and how the combination with chloroquine led to a significant induction of lysosomal membrane permeabilization coupled to massive cell death.

The manuscript is well written and organized appropriately; the large amount of experiments is consistent with the conclusions suggested by the authors.

We thank the reviewer for the positive comments.

Minor point:

I suggest to add some references regarding G4 ligands to the introduction (line 86), for examples: Am. Chem. Soc., 2007, 129 (6), 1502-1503 DOI: 10.1021/ja065591t; Nucleic Acids Research, 2012, Vol. 40, No. 12 doi:10.1093/nar/gks152; Phys.Chem.Chem.Phys., 2017, 19, 17404 DOI: 10.1039/c7cp02816d

All three references have been added in the Introduction.

The structure of 20A showed in figure 5a should be reported at the end of introduction in a new figure

As requested by the reviewer, we modified figure 5a and presented the structure of 20A and its derivatives in a new figure (Figure 1).

The authors define 20A as G4 ligand belonging to the triarylpyridine family (as demonstrated previously). However, in this manuscript this information is not so relevant to appear in the title. The authors did not demonstrate that the effect of 20A on the lysosome is due to the interaction with G4. Therefore I suggest to replace the "G-quadruplex ligands..." in the title with "triarylpyridine derivative...."

This remark is in line with the comments from Reviewer #1. We replaced "G-quadruplex ligands..." in the title with "triarylpyridine compounds”.

Round 2

Reviewer 1 Report

I am ok with the changes provided by the authors. I think it is a nice work.

Reviewer 2 Report

In this revised manuscript and the accompanying letter the authors have satisfactorily replied to most of the requests presented in the first round of review. The authors have also updated the manuscript with the most important information needed to further explain or support their data and conclusions. Since it is understandable that limitations to the laboratory routine apply by these times, I think that the manuscript does not need any further update. I have no other questions or requests for the authors.